# Mortality Prediction in Patients with Severe Acute Kidney Injury Requiring Renal Replacement Therapy

**DOI:** 10.3390/medicina57101076

**Published:** 2021-10-09

**Authors:** Žilvinas Paškevičius, Inga Skarupskienė, Vilma Balčiuvienė, Eglė Dalinkevičienė, Neda Kušleikaitė-Pere, Kristina Petrulienė, Edita Žiginskienė

**Affiliations:** 1Department of Nephrology, Medical Academy, Lithuanian University of Health Sciences, Eivenių 2, LT-50161 Kaunas, Lithuania; inga.skarupskiene@kaunoklinikos.lt (I.S.); egle.dalinkeviciene@kaunoklinikos.lt (E.D.); neda.kusleikaite@kaunoklinikos.lt (N.K.-P.); kristina.petruliene@kaunoklinikos.lt (K.P.); edita.ziginskiene@kaunoklinikos.lt (E.Ž.); 2Hospital of Lithuanian University of Health Sciences, Eivenių 2, LT-50161 Kaunas, Lithuania; vilma.balciuviene@kaunoklinikos.lt

**Keywords:** acute kidney injury, renal replacement therapy, mortality prediction

## Abstract

*Background and Objective*: Acute kidney injury (AKI) remains a serious health condition around the world, and is related to high morbidity, mortality, longer hospitalization duration and worse long-term outcomes. The aim of our study was to estimate the significant related factors for poor outcomes of patients with severe AKI requiring renal replacement therapy (RRT). *Materials and Methods*: We retrospectively analyzed data from patients (*n* = 573) with severe AKI requiring RRT within a 5-year period and analyzed the outcomes on discharge from the hospital. We also compared the clinical data of the surviving and non-surviving patients and examined possible related factors for poor patient outcomes. Logistic regression was used to analyze the odds ratio for patient mortality and its related factors. *Results:* In 32.5% (*n* = 186) of the patients, the renal function improved and RRT was stopped, 51.7% (*n* = 296) of the patients died, and 15.9% (*n* = 91) of the patients remained dialysis-dependent on the day of discharge from the hospital. During the period of 5 years, the outcomes of the investigated patients did not change statistically significantly. Administration of vasopressors, aminoglycosides, sepsis, pulmonary edema, oliguria, artificial pulmonary ventilation (APV), patient age ≥ 65 y, renal cause of AKI, AKI after cardiac surgery, a combination of two or more RRT methods, dysfunction of three or more organs, systolic blood pressure (BP) ≤ 120 mmHg, diastolic BP ≤ 65 mmHg, and Sequential Organ Failure Assessment (SOFA) score on the day of the first RRT procedure ≥ 7.5 were related factors for lethal patient outcome. *Conclusions:* The mortality rate among patients with severe AKI requiring RRT is very high—52%. Patient death was significantly predicted by the causes of AKI (sepsis, cardiac surgery), clinical course (oliguria, pulmonary edema, hypotension, acidosis, lesion of other organs) and the need for a continuous renal replacement therapy.

## 1. Introduction

For a long time, acute kidney injury (AKI) has been recognized as a severe health disorder. At present, AKI is still frequent around the world and is related to high morbidity, mortality, longer hospitalization duration and worse long-term outcomes, such as chronic kidney disease (CKD) [1,2]. Despite improvements in the quality of medical care, the incidence and mortality of AKI continues to rise [3]. In spite of the progress made in medicine, the number of AKI cases has increased, especially among hospitalized patients and in patients in intensive care units (ICU). The incidence of AKI is largely dependent on the definition used, investigated patient population and geographic region. The rates of AKI in hospitalized patients have been reported to be between 5% and 15% [4,5,6,7] with a much higher incidence in critically ill patients [8]. Severe AKI requiring renal replacement therapy (RRT) occurs in 2–7% of all ICU patients [9].

Every year, about two million people in the whole world die due to AKI, whereas survivors have an increased risk of CKD and end-stage kidney disease [10,11]. Such high mortality has already persisted for several decades, and necessitates the search for possibilities to reduce it. The aim of our study was to estimate the significant related factors for poor outcomes of patients with severe AKI requiring RRT.

## 2. Materials and Methods

We performed a retrospective analysis of all patients (*n* = 573) with severe AKI requiring RRT (AKI stage 3) in the Hospital of the Lithuanian University of Health Sciences Kauno Klinikos within a 5-year period. Analyzing our data, we used the Acute Kidney Injury Network (AKIN) criteria, and the patients who received RRT were classified as AKIN3. The standardized criteria required to start RRT in cases of severe AKI were clinical symptoms of uremia, hypervolemia, hyperkalemia, metabolic acidosis, high levels of serum creatinine (>600 µmol/L) and urea (>30 mmol/L). Hemodynamically stable patients with severe AKI were treated with intermittent hemodialysis, and continuous RRT methods were used for hemodynamically unstable patients (use of vasopressors to support hemodynamics). Decisions to initiate RRT and choose modalities of it were made only by nephrologists.

We analyzed the following outcomes of the investigated patients: mortality, improvement of renal function, and remaining dependence on dialysis on discharge from hospital. We compared the clinical data of the surviving and non-surviving patients. We examined possible related factors for poor patient outcomes: age, arterial hypotension, oliguria, pulmonary edema, administration of vasopressors and aminoglycosides, modality and duration of RRT, sepsis, causes of AKI, Sequential Organ Failure Assessment (SOFA) as an organ failure score, the presence of multiple organ dysfunction syndrome (MODS), and the application of artificial pulmonary ventilation (APV) at the time of the initiation of RRT (on the day of the first RRT procedure).

The study was approved by the Kaunas Regional Biomedical Research Ethics Committee, and informed consent was waived due to the retrospective nature of the study.

The analysis of the data was performed with the Statistical Package for Social Sciences software (version 22, SPSS, Inc., Chicago, IL, USA). The averages of parametric values (x ± SD) were calculated. Employing the Kolmogorov–Smirnov test, the distribution of quantitative values was established. Comparing the quantitative values which do not satisfy the laws of normal distribution, comparison tests of non-parametric values were performed. The difference between two independent groups was established by employing the Mann–Whitney-Wilcoxon test. The relationship between the qualitative values was evaluated following the criteria of Pearson’s chi square test. The difference between the compared groups is statistically significant when *p* < 0.05.

During the logistic regressive analysis, the odds ratio for patient mortality was established. For the most significant quantitative related factors selected according to logistic regression analysis following the median and receiver operating curves (ROC), the critical values important for patient mortality were established. For the establishment of the prognostic value of the ROC test, the area under the curve was calculated.

The mortality of patients was analyzed by employing unifactorial logistic regression analysis. The method of logistic regression has enabled researchers to establish the influence of features on the chances of the appearance of binomial events. The variables which in the comparative analysis differed statistically significantly (*p* < 0.05) were involved in the multifactorial logistic regression analysis. When the quadrate of determination coefficient was higher or equal to 0.25, the model was recognized as suitable.

## 3. Results

### 3.1. Demographic and Clinical Characteristics of Patients

In total, 573 patients (198 women (34.6%) and 375 men (65.4%)) were included in the study. The demographic and clinical characteristics of the investigated patients (*n* = 573) are summarized in Table 1.

The causes of severe AKI were renal (*n* = 180, 31.4%), prerenal (*n* = 144, 25.1%), and postrenal (*n* = 44, 7.7%), and 26.2% (*n* = 150) of the patients had mixed etiology. In 9.6% (*n* = 55) of the cases, the causes of severe AKI remained unspecified. Overall, the most frequent cause was acute tubular necrosis (27.8% of all the reasons). Severe AKI mostly developed in patients with sepsis, hepatorenal syndrome (*n* = 22, 3.8%), after cardiac surgery (*n* = 39, 6.8%), other surgical operations (*n* = 31, 5.4%) and polytrauma (*n* = 12, 2.1%).

As many as 62% (*n* = 355) of all severe AKI cases were complicated with MODS. Overall, 14.3% (*n* = 82) of all patients had two-organ dysfunction, 31.9% (*n* = 183) had three-organ dysfunction, and 15.7% (*n* = 90) of patients had complications with four-organ dysfunction.

### 3.2. Renal Replacement Therapy

In total, 573 patients underwent 3807 RRT procedures during the study period. For 75.7% (*n* = 434) of the study patients, only one RRT method was applied, for 56.7% of patients (*n* = 325), only intermittent hemodialysis (HD) was applied, for 12.9% (*n* = 74), continuous RRT (continuous venovenous hemofiltration (CVVHF) or continuous venovenous hemodiafiltration (CVVHDF)) was applied, for 5.8% (*n* = 33), slow HD was applied, and for 0.3% (*n* = 2), isolated ultrafiltration (UF) was applied. For 24.3% (*n* = 139) of the investigated patients, a combination of two or more RRT methods was applied. We analyzed differences in mortality according to the different methods of RRT. The highest mortality was in the groups of CVVHF and CVVHDF (90%) and the lowest was in intermittent HD group—30.5%. The risk of dying was higher in the CVVHDF/CVVHF group (OR = 7.730 (95% CI 3.757–15.903); *p* < 0.001) than in other RRT methods Patients with a combination of two or more RRT methods had a higher mortality risk (OR = 3.410 (95% CI 2.238–5.196); *p* < 0.001) than those subjected to one RRT method.

### 3.3. Outcomes of Severe AKI

We analyzed the outcomes of severe AKI. In 32.5% (*n* = 186) of the patients, the renal function improved and RRT was stopped, 51.7% (*n* = 296) of the patients died, and 15.9% (*n* = 91) of the patients remained dialysis-dependent at hospital discharge. During the period of 5 years, the outcomes of the investigated patients did not change statistically significantly. We analyzed the demographic and clinical data of the surviving and non-surviving patients (Table 2).

There was no statistically significant difference according gender, age and the number of hospitalization days between the groups of the survivors and non-survivors. The RRT duration of the non-survivors was statistically significantly shorter (*p* = 0.035). Mortality of the patients older than 80 (59.3%) was statistically significantly higher than that of the younger patients up to 40 (38.3%) (*p* = 0.02). The mortality of patients older than 85 reached as high as 70.6%. The mortality was strongly associated with older age (>85 y) (OD = 2.356 (95% CI 1.105–5.021); *p* = 0.026) and lower blood pH (pH < 7.3) (OD = 1.739 (95% CI 0.852–2.942); *p* = 0.008).

### 3.4. Mortality According Cause of AKI

The results show that the lowest mortality rate was in the postrenal AKI group (*n* = 6, 13.6%) and the highest was in the patient group of renal causes—62.2% (*n* = 112). Mortality rate in AKI with the mixed etiology group was 50.7% (*n* = 76), in prerenal—55.6% (*n* = 80), and in the group of unspecified causes, 40% (*n* = 22) of patients died. The mortality rate of hepatorenal severe AKI (*n* = 20) was 90.9%, while after cardiac surgery (*n* = 30) it was 76.9% and under sepsis (*n* = 91) it was 74.6%. The mortality rate of patients with severe AKI after cardiac surgery was statistically significantly higher than with severe AKI after other operations (accordingly 76.9% and 54.8%, *p* < 0.05). The renal AKI cause and septic AKI increased the risk of dying comparing with other causes, respectively (OR = 1.871 (95% CI 1.304–2.683); *p* = 0.001 and OD = 3.523 (95 % CI 2.251–5.512); *p* = 0.002).

### 3.5. Number of Affected Organs and Mortality

The results show that when the number of affected organs increased from two to four, the mortality of the patients increased significantly, from 42.7% to 92.2%. The mortality rate in the group of non-surviving patients with only kidney dysfunction was 10.2% (*n* = 30), in patients suffering from lesions in two organs it was 11.8% (*n* = 35), in patients suffering from lesions in three organs it was 50% (*n* = 148), and in patients suffering from lesions in four organs, it was 28% (*n* = 83). Overall, 87% (*n* = 216) of the patients whose kidney function was disordered alone survived, and only 13% (*n* = 28) died, whereas those under concurrent pulmonary lesion died at a rate of 79.5% (*n* = 240), for those under concurrent cardiac lesion, 82.2% (*n* = 236) died, and for those under concurrent liver lesion, 80.6% (*n* = 104) of the investigated patients died. Patients suffering from three or more organ lesions had a higher risk to dying (OR = 19.885 (95% CI 12.956–30.519); *p* < 0.001) than patients with a lower number of organ lesions.

### 3.6. Other Characteristics and Mortality

Univariate analysis showed that several characteristics differed significantly between surviving and non-surviving patients. Administration of vasopressors (OR 17.164, 95% CI; 11.306–26.056, *p* < 0.001), administration of aminoglycosides (OR 2.402, 95% CI; 1.345–4.291, *p* = 0.003), sepsis (OR 3.048, 95% CI 2.128–4.367, *p* < 0.001), pulmonary edema (PE) (OR 2.018, 95% CI; 1.391–2.926, *p* < 0.001), oliguria (OR 3.504, 95% CI; 2.430–5.054, *p* < 0.001), APV (OR 7.732, 95% CI 5.268–11.350, *p* < 0.001), patient age ≥ 65 yrs. (OR 2.356, 95% CI; 1.105–5.021, *p* = 0.026), renal cause of AKI (OR 1.871, 95% CI; 1.304–2.683, *p* = 0.001), AKI after cardiac surgery (OR 3.358, 95% CI; 1.564–7.210, *p* = 0.002), the combination of two or more RRT methods (OR 3.410, 95% CI 2.238–5.196, *p* < 0.001), dysfunction of three or more organs (OR 19.885, 95%; CI 12.956–30.519, *p* < 0.001), systolic BP ≤ 120 mmHg (OR 8.452, 95% CI; 5.784–12.352, *p* < 0.001), diastolic BP ≤ 65 mmHg (OR 5.892, 95% CI; 4.099–8.470, *p* < 0.001), and SOFA score ≥ 7.5 (OR 13.149, 95% CI 8.812–19.619, *p* < 0.001) were all related factors for lethal patient outcome. The results of two logistic regression models for patient mortality are shown in Table 3 and Table 4.

The lethal outcome was significantly associated with patient age ≥ 65, systolic BP ≤ 120 mmHg, multiple organ dysfunction syndrome (≥3-organ dysfunction), a renal cause of AKI, AKI which had developed after cardiac surgery, pH < 7.3, oliguria, and the administration of aminoglycosides and vasopressors.

## 4. Discussion

AKI is a serious clinical condition which carries high mortality and morbidity risk. Early diagnosis of AKI could lead to accurate evaluation of the case and might help to establish the prognosis and improve outcomes [12].

Age is one of the most important non-modifiable related factors for AKI. Investigators in Denmark [13] tried to show age changes in patients with AKI treated with RRT from 2000–2012. Their results showed that mean age rose continuously during this period of time: in 2000–2003, it was 64.1 ± 14.1 years, in 2004–2008, it was 66.1 ± 13.5 years, and in 2009–2012, it reached 66.8 ± 13.5 years. Our study also demonstrates that the mean age of RRT-requiring AKI patients rose from 62.8 ± 16.0 to 67.6 ± 15.1 years during the 5-year period, and the age averages are very similar in our study too. Meanwhile, scientists from southwest Nigeria [14] published completely different results. In 91 patients with AKI, the average age was 45.1 ± 20.7 years; moreover, 75.8% of patients were younger than 65 years. In our study, 47.3% of patients were under 65 years old, and that percentage dropped down to 35.8% during the 5-year period. The differences in these results could be explained by the different levels of development of the countries and healthcare centers in them.

Over many years, three main categories of AKI etiology have been established: prerenal, renal and postrenal. Yang et al. [15], in their retrospective study, analyzed clinical data and the prognosis of 271 patients with AKI. The causes of AKI were distributed as follows: 46.5% of cases AKI were renal, 36.5% were prerenal and 17% were postrenal. Their results showed that postrenal AKI was statistically significant in relation to renal function improvement compared with the other two groups of AKI causes. In our study, renal causes were also the most common (*n* = 180, 31.4%), but renal and postrenal were much less common compared with Yang et al.’s study. Additionally, in contrast to Yang et al., our results show that the highest mortality rate was in the renal AKI group. The reason for this distinction is the different severities of AKI: in the previous study, all stages of AKI were included and only 22.5% of the study group required RRT treatment. Meanwhile, in our work, we focused exclusively on RRT-requiring AKI patients.

The majority of published studies have focused on the entire spectrum of AKI severity, and only a few have exclusively focused on RRT-requiring AKI cases. Corte et al.’s study from Belgium [2] is one of these. A total of 1292 ICU patients from 2004 through 2012 with RRT-requiring AKI were included. Despite the fact that in this study, patients required treatment in the ICU and potentially were in a more severe condition, short-term mortality results were very similar to our study—51.3% and 54.6%, respectively. Moreover, a very similar proportion of the patients remained RRT-dependent at the time of discharge from hospital—13.8% in the previously mentioned study and 15.9% in our results. Another retrospective cohort study by Rennie et al. [16] also confirmed that RRT-requiring AKI is associated with longer hospitalization length, higher mortality and long-term loss of kidney function. Mortality in hospital in this study was lower (36%) compared to our study. Additionally, the reasons for that difference might be related to patient selection, different competence of health professionals, diagnostic and treatment methods, and economic and medical insurance differences.

Since mortality rates are notably high in all published studies, evaluation of the related factors for poor outcomes becomes extremely important. Luo et al. [17] in their study compared prognostic parameters between survival and non-survival groups in patients with AKI within 90 days after AKI diagnosis. They found statistical differences between two groups in characteristics such as age, AKI types, causes of AKI, mechanical ventilation, hypotension, shock, heart, respiratory, digestive and central nervous system failures. A similar study was performed in Beijing (China) [18], where analysis revealed a few more factors such as body mass index (BMI), low cardiac output, serum albumin level, leucocyte and platelet count. Meanwhile, Walker et al. [19] tried to establish related factors in recurrent AKI. As in other studies, the results show that increasing age, cardiovascular and cerebrovascular diseases were associated with higher mortality rate. By looking at our research, we can see similar prognostic parameters for negative outcomes as in multiple published studies: administration of vasopressors and aminoglycosides, application of APV, oliguria, pulmonary edema, lower systolic and diastolic blood pressures, and higher SOFA score at the first day of RRT.

One of the biggest related factors for poor patient outcomes in our study was a high SOFA score. Lee et al. [20] in their study predicted the development of septic AKI and in-hospital mortality by analyzing a combination of SOFA score and various biomarkers. In their study, 22 of 33 patients had a diagnosis of sepsis with varying degrees of AKI. Their results showed very high values in predicting in-hospital mortality in the sepsis-induced AKI group. Our study demonstrates that the proportion of patients with a diagnosis of septic AKI increased from 17.3 to 24.6% during the 5-year period. Additionally, in this group, we observed statistically significant differences in the parameters affecting severity: lower blood pressure, more common administration of vasopressors and application of APV, higher rates of PE and higher SOFA scores on the first day of RRT.

Another important subject is perioperative AKI. As in our work, there are many studies focusing on mortality rates after cardiac surgery. Systemic review and meta-analysis by Hu et al. [21] demonstrated that among 59 studies, AKI-associated mortality rates after cardiac surgery were between 10.7 and 30.0%. Meanwhile, the main investigation object of Yu Pan et al. [22] was the incidence of AKI in non-cardiovascular surgeries. In 3468 cases of hospital-acquired AKI, 30.5% of cases were determined as AKI after non-cardiovascular surgery. In our study, cardiac surgery-associated AKI comprised 6.8% of all AKI cases, and it led to more severe patient conditions: administration of vasopressors, application of APV, lower levels of arterial blood pressure, and higher values of SOFA score on the day of the first RRT procedure were statistically significant related factors. Additionally, our study demonstrated an over two times higher mortality ratio after cardiac surgery (76.9%) compared to the review and meta-analysis of Hu et al. Therefore, there is no doubt that cardiac surgery, as well as surgery in general, is a factor associated with a higher risk of negative outcomes in AKI patients.

Related factors which influence patient prognosis should be investigated further to help to identify and treat AKI in the early stages. This would help to reduce the occurrence and negative outcomes of AKI. Recent years numbers of publications about prevention and early diagnostics increased dramatically. Such innovations as automated electronic alerts, clinical decision support and information systems and measurement of various biomarkers show promising results in improvement outcomes of AKI [23,24]. So early mortality prediction for AKI patients can help health care providers to evaluate patients’ conditions and take adequate actions in time [25].

## 5. Conclusions

The most frequent outcome of severe AKI patients was death (about half of the investigated patients died); for only one third of the patients, the kidney function improved (the demand for RRT disappeared), while for 16%, the demand for RRT remained after in-patient treatment. The administration of vasopressors and APV, the administration of aminoglycosides, sepsis, oliguria, pulmonary oedema, systolic BP ≤ 120 mmHg, diastolic BP ≤ 65 mmHg, SOFA score ≥ 7.5 on the day of the first RRT procedure, sodium > 140 mmol/L, and pH < 7.3 all significantly increased the patients’ risk of dying.

The causes of acute kidney injury (sepsis, cardiac operations), clinical course (oliguria, pulmonary edema, hypotension, acidosis, lesion of other organs) and the need for a continuous RRT were prognostic factors of patient death.

Lethal outcomes of patients were significantly predicted by age (≥ 65), systolic BP ≤ 120 mmHg, multiple organ dysfunction syndrome (≥ 3-organ dysfunction), renal cause of AKI, AKI which had developed after cardiac surgery, pH < 7.3, oliguria, and the administration of aminoglycosides and vasopressors.

Intense concern about pure outcomes of severe AKI is increasing despite an adequate RRT. That is why early mortality prediction is critically important for AKI patients. So, attention of health care providers needs to be focused on early establishment and correction of AKI risk factors and interventional treatment of AKI at the appropriate time.

## Figures and Tables

**Table 1 medicina-57-01076-t001:** Demographic and clinical characteristics of study patients.

Characteristics of Patients	Units	Value
Gender: men/women	*n* (%)	375 (65.4)/198 (34.6)
Age	median(25–75%), years	69 (56–77)
Length of hospitalization	median (25–75%), days	17 (7–36)
Length of hospitalization till getting into ICU	median (25–75%), days	6 (2–14)
Length of hospitalization in ICU till the beginning of RRT	median (25–75%), days	2 (1–6)
CKD (the 2nd–5th CKD stage)	*n* (%)	171 (29.8)
CKD (the 1st CKD stage)	*n* (%)	23 (4.0)
Administration of vasopressors	*n* (%)	275 (48)
Application of APV	*n* (%)	310 (54.1)
Sepsis without a septic shock	*n* (%)	48 (8.4)
Sepsis with a septic shock	*n* (%)	160 (27.9)
At the time of the initiation of RRT:		
Pulmonary oedema	*n* (%)	166 (29.0)
Systolic BP	mean ± SD, mmHg	120.4 ± 28.9
Diastolic BP	mean ± SD, mmHg	66 ± 16.6
SOFA score	mean ± SD	8.4 ± 3.9
pH	mean ± SD	7.3 ± 0.1
Bicarbonate (HCO_3_)	mean ± SD, mmol/L	15.4 ± 5.5
RRT duration	median (25–75%), days	4 (2–14)
RRT duration	median (25–75%), hours	16 (6–42)

ICU—intensive care unit, RRT—renal replacement therapy, CKD—chronic kidney disease, APV—artificial pulmonary ventilation, BP—arterial blood pressure.

**Table 2 medicina-57-01076-t002:** The demographic and clinical data of the survivors and non-survivors with severe acute kidney injury.

Characteristics of Patients	Survivors (*n* = 277)	Non-Survivors (*n* = 296)	*p* Value
Gender, n (%): men/women	180 (48)/ 97 (49)	195 (52)/ 101 (51)	0.137/ 0.344
Age (mean ± SD), years	64.5 ± 15.9	66.3 ± 15.5	0.153
Length of hospitalization till getting into ICU (mean ± SD), days	12.4 ± 17.1	9.5 ± 12.6	0.165
Length of stay in ICU till the beginning of RRT (mean ± SD), days	4.5 ± 5.2	4.9 ± 7.5	0.697
CKD (the 2nd–5th CKD stage), n (%)	102 (36.8)	69 (23.3)	0.006
CKD (the 1st CKD stage), n (%)	10 (3.6)	13 (4.4)	0.317
Administration of vasopressors, n (%)	46 (16.6)	229 (77.4)	<0.001
Administration of aminoglycosides, n (%)	18 (6.8)	42 (15)	0.002
Sepsis n (%)	65 (23.5)	143 (48.3)	<0.001
At the time of the initiation of RRT:			
Pulmonary oedema, n (%)	60 (21.7)	106 (35.8)	<0.001
Oliguria, n (%)	139 (50.9)	229 (78.4)	<0.001
APV, n (%)	53 (19.3)	191 (65)	<0.001
Systolic BP (mean ± SD), mmHg	134.6 ± 27.9	107 ± 22.9	<0.001
Diastolic BP (mean ± SD), mmHg	73.5 ± 14.7	58.8 ± 15.2	<0.001
SOFA score (mean ± SD)	6.1 ± 2.5	10.5 ± 3.9	<0.001
pH (mean ± SD)	7.3 ± 0.1	7.6 ± 0.1	0.027

ICU—intensive care unit, RRT—renal replacement therapy, CKD—chronic kidney disease, APV—artificial pulmonary ventilation, BP—arterial blood pressure.

**Table 3 medicina-57-01076-t003:** Associations of variables and mortality of patients with dialysis-dependent acute kidney injury in the model of multivariable logistic regression analysis. (Model 1)

Factors	OR of Mortality (95% CI)
Age ≥ 65 years	1.722 (1.08–2.746)
Systolic BP ≤ 120 mmHg	3.336 (2.073–5.368)
Oliguria	2.226 (1.373–3.608)
Multiple organ dysfunction syndrome (≥3-organ dysfunction)	6.26 (3.18–12.322)
Administration of vasopressors	2.354 (1.186–4.669)

Constant = −2.676, *p* < 0.001. The model correctly forecasts 82.1%, Nagelkerke R Square = 0.549, OR—odds ratio, CI—confidence interval.

**Table 4 medicina-57-01076-t004:** Associations of variables and mortality of patients with dialysis-dependent acute kidney injury in the model of multivariable logistic regression analysis. (Model 2)

Factors	OR of Mortality (95% CI)
Renal AKI cause	1.679 (1.047–2.694)
AKI after cardiac surgery	3.640 (1.453–9.118)
pH < 7.3	1.812 (1.146–2.866)
Administration of aminoglycosides	2.447 (1.149–5.209)
Oliguria	4.263 (2.586–7.028)

Constant = −1.332, *p* < 0.001. The model correctly forecasts 69.8%, Nagelkerke R square = 0.203, OR—odds ratio, CI—confidence interval.

## Data Availability

Not applicable.

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
