# Peer review of "Mortality Prediction in Patients with Severe Acute Kidney Injury Requiring Renal Replacement Therapy"

_medicina, 2021, doi:10.3390/medicina57101076_

Round 1
Reviewer 1 Report
The authors investigated the association between acute kidney injury (AKI) and mortality in an intensive care unit.
The topic is crucial in clinical practice; however, there have been a lot of studies focusing on AKI. Therefore, I think that this study lacked novelty.
Nonetheless, the number of patients who were included in this study was large, and this study may partially contribute to the treatment of AKI.
There are several concerns in this study.
Major points
- The authors used the term “risk factors” throughout the manuscript; however, they did not use the time from developing AKI to death. Generally, it is impossible to elucidate risk using logistic regression models. Therefore, the authors should not use the term “risk factors” from the result of this study.
- I guess that logistic regression models were conducted by stepwise method to augment the value of the R square. It would be better for the author to show how they select the parameters in the logistic regression models. In addition, the number of patients who died of AKI, and I want to know the result from a logistic regression model that contains a lot of parameters or patients’ backgrounds.
- Hepatorenal AKI is a very severe condition, and liver transplantation or simultaneous liver and renal transplantation are considered to save the patients’ lives. Were there any patients who underwent liver transplantation?
Minor points
- The style of showing the value is not good throughout the table. For example, systolic BP at the initiation of RRT; 120.38±28.9 should be 120.4±28.9.
- Were decisions initiating renal replacement therapy or choosing modalities made by more than one person? If so, please state the fact.
Author Response
Dear reviewer,
We sincerely appreciate all your comments and suggestions. We are grateful for understanding the importance of this topic and acknowledgement of possibility of contribution of our study in treatment of AKI in the future.
The majority of published studies have focused on the entire spectrum of AKI severity (AKI stage 1-3). But our focus was exclusively on AKI requiring RRT (AKI stage 3). That is uniqueness of our study. Also this is the first study about third stage AKI in our country with such a large study group. With this article we hope to lay the foundation for further studies about RRT required AKI in Lithuania and others smaller countries.
Now we provide response to all of your comments point-by-point.
First (major) point: We gratefully accept your first comment about the term „risk factors“ . All authors agreed with the fact that this term might be not completely correct in our case. So we decided to change it into „related factors“.
Second (major) point: At first we established the odds ratio for patient mortality by using logistic regressive analysis. After the evaluation of all possible related factors (all of them are in table 2), by using univariate analysis we isolated only statistically significant quantitative related factors. Then we established the critical values important for patient mortality. We decided to not mention nonsignificant related factors in further text of the article with the purpose of keeping the volume of the manuscript smaller.
Third (major) point: There wasn‘t any patients who underwent liver transplantation in our study. And the reason of not absolute mortality of patients with hepatorenal syndrome might be related to not fully correct diagnosis of HRS by the clinicists.
First (minor) point: Thank you for this notice, we corrected the values in entire manuscript.
Second (minor) point: Decisions to initiate RRT and choose modalities of it were made by more than one person, but all them were nephrologists. Indications to initiate RRT are standardized in our hospital. We stated that fact in the „Materials and methods“ part of the manuscript.
Thank you again for all of your thoughts.
Kind regards,
Mr Žilvinas Paškevičius

Reviewer 2 Report
colleagues " ". analysis outcomes at .(i.e. see Jurawan, N., Pankhurst, T., Ferro, C. et al. Hospital acquired Acute Kidney Injury is associated with increased mortality but not increased readmission rates in a UK acute hospital. BMC Nephrol 18, 317 (2017). https://doi.org/10.1186/s12882-017-0729-9 and Pan Y, Wang W, Wang J, Yang L, Ding F; ISN AKF 0by25 China Consortium. Incidence and Risk Factors of in-hospital mortality from AKI after non-cardiovascular operation: A nationwide Survey in China. Sci Rep. 2017 Oct 24;7(1):13953. doi: 10.1038/s41598-017-13763-9).
Nevertheless, .
written, .
Author Response
Dear reviewer,
We sincerely appreciate all your comments and suggestions. We are grateful for understanding the importance of this topic and acknowledgement of possibility of contribution of our study in treatment of AKI in the future.
We are very grateful for all the good words about our manuscript from you and as you suggested we extended the part of clinical relevance of our research in the conclusions.
Kind regards,
Mr Žilvinas Paškevičius

Round 2
Reviewer 1 Report
I do not have any comments. The authors responded and revised the manuscript properly.